# Gene polymorphisms of Patatin-like phospholipase domain containing 3 (PNPLA3), adiponectin, leptin in diabetic obese patients

Omnia Aly[1], Hanan Hassan Zaki[1], Mohamed R. Herzalla[2], Ahmed Fathy[2], Nermin Raafat[3], Mohamed M. Hafez[4]*

1 Department of Medical Biochemistry, National Research Centre, Cairo, Egypt, 2 Department of Internal Medicine, Faculty of Medicine, Zagazig University, Zagazig, Egypt, 3 Medical Biochemistry Department, Faculty of Medicine, Zagazig University, Zagazig, Egypt, 4 Biochemistry Department, Faculty of Pharmacy, Ahram Canadian University, 6th of October City, Egypt

* mohkhalifa2000@yahoo.com

**Data Availability Statement:** All relevant data are within the paper and its Supporting Information files.

## Abstract

Obesity leads a crucial importance in metabolic disorders, as well as type 2 diabetes mellitus. Our present study was designed to assess the potential role of irisin, adiponectin, leptin and gene polymorphism of PNPLA3, leptin and adiponectin as predictive markers of diabetes associated with obesity. One hundred eighty subjects were distributed to three groups including; healthy non-diabetic non obese volunteers as a control group, diabetic non obese group, and diabetic obese group (n = 60 for each group). Fasting blood samples of all groups were collected to determine fasting blood glucose, insulin levels, insulin resistance, total cholesterol, high density lipoprotein cholesterol (HDL-C), low density lipoprotein cholesterol (LDL-C), triacylglycerol, irisin, adiponectin, leptin; as well as, polymorphism of PNPLA3, adiponectin and leptin. The results showed that glucose, insulin resistance, total cholesterol, irisin, leptin, LDL-C, triacylglycerol concentrations were significantly increased, however, insulin, HDL-C, adiponectin were significantly decreased in diabetic obese patients in relation to diabetic non-obese patients as well as in healthy volunteers. The polymorphism of PNPLA3 rs738409 was linearly related to irisin and leptin but was not related with circulating concentrations of adiponectin. We concluded that increased irisin and leptin levels can predict the insulin resistance in obese patients. Moreover, patients who have mutant genotype of *PNPLA3* I148 gene (rs738409) C>G, ADIPOQ gene (rs266729) G>C and LEP gene (rs2167270) G>A showed a significant higher susceptibility rate for DM in obese people than those with wild type. This could be considered as an adjustable retort to counter the impact of obesity on glucose homeostasis.

## 1. Introduction

Obesity has become an epidemic of environmental health throughout the world and was one of the real hazard factors for type 2 diabetes mellitus (T2DM) [1]. It is also correlated with exaggerated adipose tissue mass and adipocyte dysfunction, inflammatory manufacturing of

**Funding:** The author(s) received no specific funding for this work.

**Competing interests:** The authors have declared that no competing interests exist.

**Abbreviations:** ADIPOQ, Adiponectin; ATMs, Adipose tissue macrophages; CETP, Cholesterol ester transfer protein; FFAs, Free fatty acids; FNDC5, Fibronectin type III domain containing 5; MetS, Metabolic syndrome; PCR-RFLP, Polymerase chain reaction-restriction fragment length polymorphism; PGC-1α, Peroxisome proliferator activated receptor gamma coactivator-1 alpha; PNPLA3, Patatin-like phospholipase domain containing 3; T2DM, Type 2 diabetes mellitus.

adipokines, oxidative stress, endoplasmic reticulum stress and insulin resistance. These strategies came into contact with diabetes and obesity and led to the development of diabesity [2].

Irisin is a glycosylated polypeptide hormone in muscles as the type 1 film forerunner protein fibronectin type III domain containing 5 (FNDC5), which is proteolytically broken and formed. This protein was portrayed as a myokine-instigated practice that triggers caramelizing of white fat tissue and lifts the metabolic qualities' appearance. Moreover, it is delivered because of incitation of peroxisome proliferator activated receptor gamma coactivator-1 alpha (PGC-1α). It increases the total energy consumption of the body and improves thermogenesis. Increased oxidative metabolism and weight loss significantly improve glucose tolerance and resistance to insulin [3].

Adiponectin is a protein hormone that alters few metabolic procedures, for example, glucose control and unsaturated fat oxidation. It is constantly begun from adipocytes. Adiponectin has hostile against atherosclerosis and hostile to insulin opposition characters. It contributes to the elimination of metabolic disorders that can lead to obesity, atherosclerosis and cardiovascular risk. High adiponectin blood levels are correlated with a lower risk of heart attacks [4]. Adiponectin gene polymorphism is statistically significantly correlated with the DM danger, insulin resistance and coronary artery diseases [5].

Leptin is a hormone arising from adipocytes that involve lipid and glucose metabolism, energy consumption, adjustment of pro-inflammatory T-lymphocytes, acute anti-inflammatory and innate immune reactions [6]. Leptin concentrations are revealed to be accelerated by extra fat to reduce caloric intake by reducing appetite and by increasing energy depletion [7].

Patatin-like phospholipase domain containing 3 (PNPLA3), otherwise called adiponutrin, P.I148M polymorphism that substitutes methionine for isoleucine at residue 148 (I148M) is as often as possible considered a risk factor for both expanded fat gathering and fibrosis [8]. Different human studies have also demonstrated a possible direct or indirect impact of PNPLA3 genotype on adipose tissue biology [9–11].

Our present study aimed to evaluate the PNPLA3, leptin and adiponectin genes polymorphism and the irisin, leptin and adiponectin levels as significant specific biomarkers for obesity and insulin resistance in type 2 diabetes mellitus.

## 2. Subjects and methods

### 2.1. Subjects

One hundred and eighty subjects had participated in this study. They were distributed into three groups (each contains 60 individuals): Group (1) has apparently healthy volunteers as a control group, Group (2) has 60 diabetic non-obese patients and Group (3) contains 60 diabetic obese patients joined the outpatient clinic, Department of Internal Medicine, Zagazig University Hospital from June 2018 to July 2019.

Our study was authorized and approved by Ahram Canadian University (ACU) Human Ethics Committee (PBC 2019–01). Informed written consent was obtained from all subjects. All data/samples were fully anonymized before accessing them.

**2.1.1 Inclusion criteria.** 20 to 40 years adult old patients were included. Obese diabetic patients (chronic T2DM) were diagnosed by fasting blood glucose level $\geq$ 140 mg/dl, BMI was more than 30 kg/m$^2$. Their waist circumference was > 88 cm in female and more than > 102 in male. They assumed a sedentary lifestyle and were not athletes. None of them was taking medications that might affect their performance or their appetite (S1 Data).

**2.1.2. Exclusion criteria.** Individuals with body mass index less than 30 kg/m$^2$ were excluded. Also, if there was any other medical complications as liver diseases, renal diseases

and chronic hypertension. Furthermore, patients who were taking medications which affect their appetite as well as smokers were excluded.

## 2.2. Methods

**2.2.1. Sample collection.** Five ml venous blood samples were drawn under complete aseptic conditions and subdivided into two portions: first part is collected in EDTA tubes for DNA extraction & genotyping and the second part was drawn into plain tubes and serum was separated to identify other biochemical markers.

**2.2.2. Biochemical analysis.** *2.2.2.1. Evaluation of insulin resistance.* Fasting blood glucose levels were estimated by enzymatic colorimetric technique, using kits of Centronic Co. (Wartenberg, Germany), according to method of Trinder [12]. Serum insulin levels were evaluated by enzyme linked immunosorbent assay (ELISA) [13] using BioSource INSEASIA Co. (Nivelles, Belgium) Kit. Insulin resistance was determined from this equation: Insulin resistance = fasting glucose (mg/dl) ×fasting insulin (μIU/ml)/405, according to Mathews *et al.* [14].

*2.2.2.2. Estimation of lipid profile.* Cholesterol, HDL-C and triacylglycerol were estimated by the colorimetric enzymatic method according to Allain *et al.* [15], Lopes-Virella *et al.* [16] and Glick *et al.* [17] respectively. The kits were supplied from Biocon Diagnostic (Germany). LDL-cholesterol was determined utilizing this equation: LDL-C (mg/dl) = Total cholesterol—(HDL-C+ TG/5) according to Friedewald *et al.* [18].

*2.2.2.3. Estimation of serum irisin, adeponectin and leptin levels.* Irisin, Adiponectin and leptin levels were evaluated by ELISA [19–21] using commercial kit purchased from DRG, USA.

*2.2.2.4. Genotyping of PNPLA3 C>G p.I148M (rs738409) by polymerase chain reaction-restriction fragment length polymorphism (PCR-RFLP) according to Dutta* [22]. **2.2.2.4.1. DNA extraction**. DNA samples were isolated from peripheral blood leucocytes using DNA extraction kit (QIAGEN DNA Prep, Germany). PNPLA3 variant was genotyped by means of PCR-RFLP method [23].

**2.2.2.4.2. PNPLA3 gene amplification by PCR**. A 333-bp region of PNPLA3 gene was amplified by PCR (Fig 1) using specific primers (forward primer: 5´–TGGGCCTGAAGTCCGAGGGT–3´ and reverse primer: 5´– CCGACACCAGTGCCCTGCAG–3´). PCR mixture (25 μl) consisted of 10.5μl of ddH₂O, 2.5 μl 10xPCR buffer, 2.5 μl 25 mMMgCl₂, 2.5 μl 2.5 mMdNTP Mix, 1,5 μl (10 pkmol/ μl) of each oligonucleotide primer, 0.5ul (1.5 units.) of "hot-start" Taq-polymerase and 5μl of DNA. PCR amplification was performed in Mastercycler® nexus (Eppendorf). The PCR conditions were: 95 ˚C for 5 min, and then 37 cycles of 94˚C for 30 sec, 66˚C for 30 sec, and 72 ˚C for 40 sec and a final extension step of 72˚C for 5 minutes.

**2.2.2.4.3. PNPLA3 rs738409 genotyping**. PCR products were processed overnight at 65˚C with BstF5 I. Digested PCR products were exposed to horizontal electrophoresis (Pharmacia Biotech by SEMKO AB, Sweden) and submarine chamber (Maxicell, EC360, M-E-C apparatus Co. St Petersburg, Florida, USA), in 1.5% ethidium bromide-recolored agarose gels in 1X TBE buffer at 120 V for 1 hr and were pictured utilizing ultraviolet trans-illumination (Heralab GmbH laborgerate trans-illuminator, Germany). Interpretation of genotyping results was performed on the basis of different bands pattern: CC genotype: 200 and 133 bp, CG genotype: 333, 200 and 133 bp, GG genotype: 333 bp (Fig 2).

*2.2.2.5. Adiponectin−11377 G>C (rs266729) genotyping by real time PCR (RT-PCR).* Genotypes of ADIPOQ gene (rs266729) were determined using allelic segregation measures with TaqMan probes (Applied Biosystems, Carlsbad, California, USA). Adiponectin (ADIPOQ) rs266729 TaqMan® Pre-Designed SNP Genotyping Assay was utilized (measure IDs: C___2412786_10), including proper primers (forward primer: 5'-ccatctcctcctcacttcc-3' and reverse primer: 5'-atgaccgggcagagctaata-3') and fluorescently named MGB™ probes (FAM and

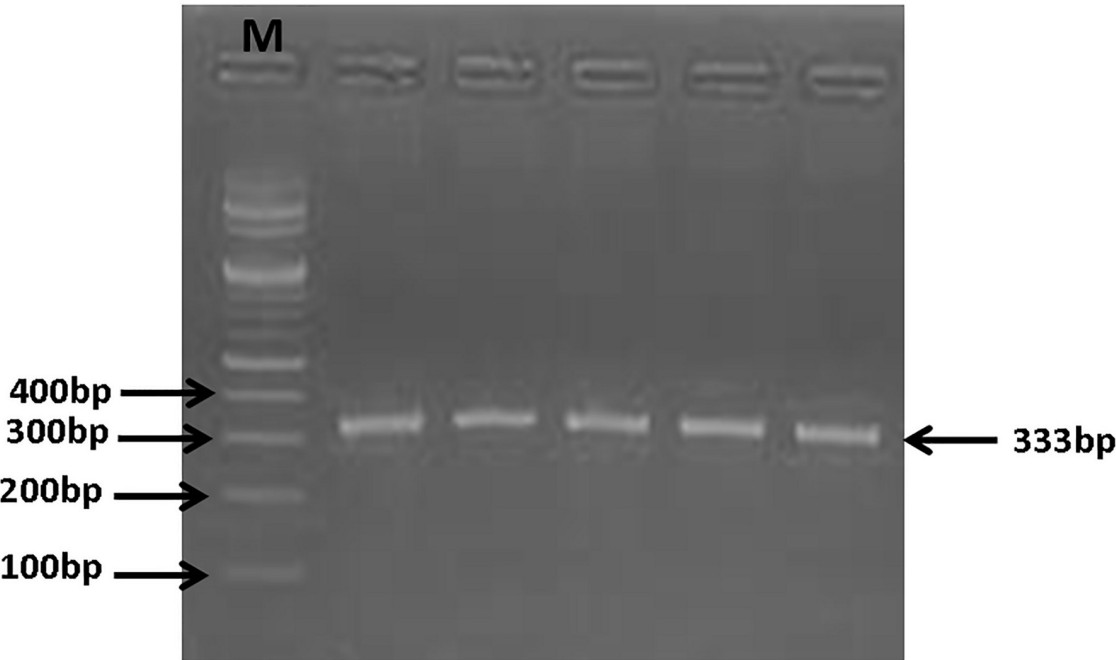

**Fig 1. Amplified PCR product of PNPLA3 gene: A representative agarose gel picture showing 333 bp bands that refer to amplified PNPLA3 gene product.** M: (100 bp—1.5 kb) DNA ladder.

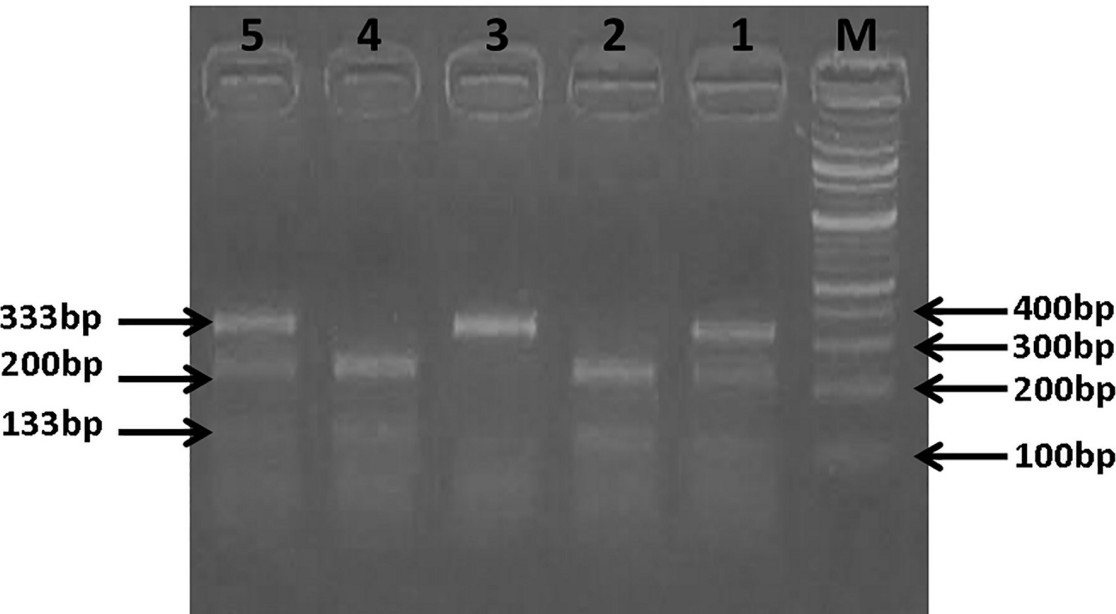

**Fig 2. Analysis of PNPLA3 gene (rs738409) C>G polymorphism: A representative agarose gel picture showing PCR-RFLP analysis of PNPLA3 gene (rs738409) C>G genotypes in genomic DNAs of study subjects with restriction endonuclease enzyme BstF5 I.** M: (100 bp—1.5 kb) DNA ladder, Lane 1 and 5: CG heterozygous (333 bp, 200 bp and 133bp), Lane 2 and 4: CC homozygous (200 bp and 133 bp) and Lane 3: GG homozygous (333 bp).

VIC) to identify the various alleles were also used. Each PCR was carried out in a total volume of 25 µl (12.5 µl of TaqMan universal PCR Master Mix, 0.625 µl of TaqMan assay, and 6.875 µl of milli-Q water and 5 µl of DNA). Thermal cycling was performed as follows: 63 ˚C for 1 min and 95 ˚C for 3 min, followed by 40 cycles at 95 ˚C for 15 s and 63 ˚C for 40 s. After PCR amplification, allelic discrimination was performed using Stratagene, MX3000P quantitative PCR System and investigated utilizing MxPro QPCR Software (Agilent innovations). As the Taq polymerase extends the primer and synthesizes the new strand, its 5' to 3' exonuclease activity degrades the probe that has annealed to the template. Degradation of the probe releases the fluorophore from it and breaks the close proximity to the quencher, thus releasing the quenching effect and allowing fluorescence of the fluorophore. Hence, fluorescence detected in the quantitative PCR thermal cycler is directly proportional to the fluorophore released and the amount of DNA template present in the PCR. Sample genotyping was automatically recognized for allelic discrimination.

*2.2.2.6. Leptin 19 G<A (rs2167270) genotyping by allele discrimination (RT-PCR).* The polymorphisms of G19A (rs2167270) in the leptin gene were investigated using the allelic discrimination method. TaqMan primers and probes were ordered as Assay-on-Demand (Assay ID: C__15966471_20) from Applied Biosystems (Carlsbad, California, USA) including appropriate primers (forward primer: 5' aaccctgtgcggattcttgtgg-3' and reverse primer: 5'-ccggtgactttctgtttggaggag-3') and fluorescently labelled (FAM and VIC) MGB™ probes to detect the alleles. The reactions were carried out using a 25 µl total volume containing 5µlof sample DNA, 12.5 µl of TaqMan universal PCR Master Mix, 0.625 µl of TaqMan assay, and 6.875 µl of milli-Q water. PCR was performed at 50 ˚C for 2 min, 95 ˚C for 10 min followed by 45 cycles at 92 ˚C for 15 s and 60 ˚C for 1 min. The reactions were then analyzed using the allelic discrimination endpoint analysis of the sequence detection on Stratagene, MX3000P quantitative PCR System and investigated utilizing MxPro QPCR Software (Agilent innovations).

## 2.3. Statistical analysis

All data was stated as mean ± SEM. The normal distribution of data was confirmed using the normal state test(SPSS package) (version 18). Statistical significance was examined by one way analysis of variance (ANOVA) trailed by Bonferroni post-hoc investigation. The genotype and allelic frequencies were evaluated by Hardy-Weinberg equilibrium (HWE) and compared by Chi-square test and Fisher's exact test, by calculating $X^2$ and 95% confidence interval (CI). P$\leq$ 0.05 was considered as statistically significant.

## 3. Results

Table 1 elucidated that blood glucose level was increased significantly by 128.77% and 300.91% in diabetic non obese and diabetic obese patients respectively when compared to control

**Table 1. Fasting blood glucose, serum insulin and insulin resistance in the different studied groups represented as mean ± SEM.**

| Parameters | Groups | | |
|---|---|---|---|
| | Group I control (n = 60) | Group II Diabetic non obese (n = 60) | Group III Diabetic Obese (n = 60) |
| **Fasting serum glucose mg/dl** | 88.3 ± 0.48 | 202 ± 0.85 [a] | 354.0 ± 1.81[a,b] |
| **Serum insulin µ IU/ml** | 16.15 ± 2.01 | 12.05 ± 1.18 [a] | 9.1 ±1.05 [a,b] |
| **Insulin resistance** | 3.52 ± 0.61 | 6.01 ±1.02 [a] | 7.95 ±1.81 [a,b] |

[a] Significant when compared to control group (P<0.05).

[b] Significant when compared to diabetic non obese group (P<0.05).

group. Furthermore, diabetic obese group showed a critical rise in blood glucose by 75.25% when compared with diabetic non obese group (P≤ 0.05).

Meanwhile, diabetic non obese patients' group showed critical decline (or sharp decrease) in serum insulin levels by 25.39% compared to normal control group. Also, diabetic obese group recorded a critical decline in it by 43.65% compared to normal control group (P≤ 0.05). Also, diabetic obese group displayed a critical decline in it by 24.48% in comparison to diabetic non obese group (P≤ 0.05) (Table 1).

Moreover, insulin resistance was increased by 70.74% and 125.85% in diabetic non obese patients and diabetic obese group respectively as compared to normal control group (P≤ 0.05). But, diabetic obese group recorded a critical decline in it by 32.28% in comparison to diabetic non obese group (P≤ 0.05) (Table 1).

Table 2 revealed that serum total cholesterol levels had a rise by 16.43% in diabetic non obese patients and by 47.06% in diabetic obese group as compared to normal control group (P≤ 0.05). Also, diabetic obese group displayed a critical total cholesterol rise by 26.31% in relation to diabetic non obese group (P≤ 0.05).

Diabetic non obese patients showed rise in serum triacylglycerol levels by 30.1% and diabetic obese group recorded a critical rise in it by 130.63% when compared with normal control group (P≤ 0.05). Likewise, diabetic obese group had 77.27% increase when compared to diabetic non obese group (P≤ 0.05) (Table 2).

Meanwhile, serum HDL-cholesterol levels were decreased by 20.32% in diabetic non obese patients' group and by 32.48% in diabetic non obese patients as compared with control group (P≤ 0.05). But, diabetic obese group displayed no critical change compared with diabetic non obese group (P≤ 0.05) (Table 2).

However, serum LDL-cholesterol levels indicated 59.29% increase in diabetic non obese patients' group and a critical rise by 130.87% in diabetic obese group in comparison to normal control group (P≤ 0.05). Whereas, diabetic obese group recorded no critical change as compared to diabetic non obese group (P≤ 0.05) (Table 2).

Table 3 revealed that diabetic non obese patients' group had53.93% rise in serum leptin levels and diabetic obese group recorded a critical rise of 109.11% as compared to control group. Additionally, comparison between diabetic obese and diabetic non obesegroupsshowed a critical rise by 35.85% (P≤ 0.05).

In the meantime, serum adiponectin levels were decreased by 41.96% and 46.4% in diabetic non obeseand diabetic obese patients respectively when compared to normal control group

**Table 2. Lipids profile levels in the different studied groups represented as mean ± SEM.**

| Parameters | Groups | | |
|---|---|---|---|
| | Group I control (n = 60) | Group II Diabetic non Obese (n = 60) | Group III Diabetic Obese (n = 60) |
| Total cholesterol mg/dL | 146.47±11.21 | 170.53±10.40[a] | 215.4±13.17 [a,b] |
| Triacylglycerol mg/dL | 50.73±5.32 | 66±5.73 [a] | 117±8.11 [a,b] |
| HDL-cholesterol mg/dL | 75.13±4.56 | 59.86±4.05 [a] | 50.73±3.64 [a] |
| LDL-cholesterol mg/dL | 61.19±5.93 | 97.47±6.39 [a] | 141.27±9.25 [a] |

[a]: Significant compared to control group (P<0.05).

[b]: Significant compared to diabetic non obese group (P<0.05).

**Table 3. Serum leptin, adiponectin and Irisin levels in the different studied groups represented as mean ± SEM.**

| Parameters | Groups | | |
|---|---|---|---|
| | Group I control (n = 60) | Group II Diabetic non Obese (n = 60) | Group III Diabetic Obese (n = 60) |
| **Leptin** ng/mL | 13.50 ± 1.80 | 20.78 ± 2.50 [a] | 28.23± 2.08 [ab] |
| **Adiponectin** ng/mL | 6.53 ± 2.42 | 3.79 ± 1.80 [a] | 3.50 ±0.84 [a] |
| **Irisin** ng/mL | 60.22±1.67 | 79.67±2.34 [a] | 94.31 ± 4.30 [ab] |

[a]: Significant compared to control group (P<0.05).

[b]: Significant compared to diabetic non obese group (P<0.05).

(P≤ 0.05). However, diabetic obese group recorded no critical change in relation to diabetic non obese group (P≤ 0.05) (Table 3).

Furthermore, diabetic non obese group indicated critical rise in serum irisin levels by 32.3% and diabetic obese group had 56.61% rise when compared to normal control group. In addition, diabetic obese group recorded a critical rise in serum irisin by 18.38% as compared to diabetic non obese group (P≤ 0.05) (Table 3).

## Polymorphism of PNPLA3 gene (rs738409)

In the control group I: Table 4 clarified that the CC genotype frequency n (%) was 26 (43.4%), the CG genotype frequency was 20 (33.3%) and the GG genotype frequency was 14 (23.3%). While in the diabetic non obese group II: the CC genotype frequency was 18 (30%), the CG genotype frequency was 14 (23.3%), and the GG genotype frequency was 28 (46.7%). Moreover, in the diabetic obese group III: the CC genotype frequency was 46 (76.67%), the CG genotype frequency was 2 (3.33%), and the GG genotype frequency was 12 (20%). There was a

**Table 4. Distribution of genotypes of PNPLA3 (rs738409) C>G, ADIPOQ rs266729 G>C and LEP rs2167270 G>A among cases and control groups.**

| Genotype | Group | | | X² | P |
|---|---|---|---|---|---|
| | Control group I | Diabetic non-Obese group II | Diabetic Obese group III | | |
| | n (%) | n (%) | n (%) | | |
| **PNPLA3 (rs738409)** | | | | | |
| **CC** | 26 (43.4) | 18 (30.0) | 46 (76.67) | 7.2 | 0.03[a] |
| **CG** | 20 (33.3) | 14 (23.3) | 2 (3.33) | 20.4 | <0.001 [b] |
| **GG** | 14 (23.3) | 28 (46.7) | 12 (20.0) | 32 | <0.001 [c] |
| **ADIPOQ (rs266729)** | | | | | |
| **CC** | 22 (36.67) | 20 (33.3) | 40 (56.7) | 5.8 | 0.05 [a] |
| **CG** | 32 (53.33) | 24 (40) | 16 (30) | 11 | 0.004 [b] |
| **GG** | 6 (9.0) | 16 (26.7) | 4 (13.3) | 15.5 | <0.001 [c] |
| **LEP (rs2167270)** | | | | | |
| **GG** | 34 (56.7) | 20 (33.3) | 18 (30) | 7.9 | 0.02 [a] |
| **GA** | 18 (30) | 22 (36.7) | 8 (13.3) | 24.9 | <0.001 [b] |
| **AA** | 8 (13.3) | 18 (30) | 34 (56.7) | 11.6 | 0.003 [c] |

[a]: p value when group II is compared to control group.

[b]: p value when group III is compared to control group.

[c]: p value when group II is compared to group III.

critical difference between diabetic non obese group II with control group I ($X^2 = 7.2$, P = 0.03). Also, There was a significant difference between diabetic obese group III with control group I ($X^2 = 20.4$, P<0.001). Besides, There was a critical difference between diabetic obese group III with diabetic non obese group II ($X^2 = 32$, P<0.001).

## Polymorphism of ADIPOQ gene (rs266729)

In the control group I: Table 4 clarified that the CC genotype frequency n (%) was 22 (36.67%), the CG genotype frequency was 32 (53.33%), and the GG genotype frequency was 6 (9%). While in the diabetic non obese group II: the CC genotype frequency was 20 (33.3%), the CG genotype frequency was 24 (40%), and the GG genotype frequency was 16 (26.7%). Furthermore, in the diabetic obese group III: the CC genotype frequency n (%) was 40 (56.7%), the CG genotype frequency was 16 (30%), and the GG genotype frequency was 4 (13.3%). There was a critical difference between diabetic non obese group II with control group I ($X^2 = 5.8$, P = 0.05). Also, There was a critical difference between diabetic obese group III with control group I ($X^2 = 11$, P = 0.004). Besides, There was a critical difference between diabetic obese group III with diabetic non obese group II ($X^2 = 15.5$, P<0.001).

## Polymorphism of LEP gene (rs2167270)

In the control group I: Table 4 clarified that the GG genotype frequency n (%) was 34 (56.7%), the GA genotype frequency was 18 (30%), and the AA genotype frequency was 8 (13.3%). While in the diabetic non obese group II: the GG genotype frequency was 20 (33.3%), the GA genotype frequency was 22 (36.7%), and the AA genotype frequency was 18 (30%). Additionally, in the diabetic obese group III: the GG genotype frequency was 18 (30%), the GA genotype frequency was 8 (13.3%), and the AA genotype frequency was 34 (56.7%). There was a critical difference between diabetic non obese group II with control group I ($X^2 = 7.9$, P = 0.02). Also, There was a critical difference between diabetic obese group III with control group I ($X^2 = 24.9$, P<0.001). Besides, There was a critical difference between diabetic obese group III with diabetic non obese group II ($X^2 = 11.6$, P = 0.003).

## 4. Discussion

The American Diabetes Association describes Diabetes mellitus as a group of degenerative disorders that are demonstrated by hyperglycemia induced by insulin secretion disorders, insulin action or both [24].

T2DM is described by insulin resistance mainly due to modified insulin production however with a certain capacity for insulin production without destroying autoimmune β-cell. Being overweight, obesity and age are possible disorders for this diabetes form. T2DM is usually undiagnosed, meanwhile hyperglycemia occurs slowly and sometimes symptomless, and ketoacidosis rarely occurs [25].

Obesity is a common disorder that has raised doubts all over the world. World Health Organization (WHO) clarified that, about 2.3 billion obese individuals who are 15 years and above, and over 700 million obese ones worldwide were present in 2015 [26]. Overweight and obesity are the key drivers of co-morbidity, along with T2DM, coronary heart disorders, oncogenes and a lot of complications that may increase morbidity and death [27].

Our work was undertaken to emphasize the role of glucose, insulin, insulin resistance, irisin, adiponectin, leptin, lipid profile and PNPLA3, leptin and adiponectin genes polymorphism as significant biomarkers in diabetic obese patients.

In this study, there was a critical rise in serum glucose concentrations in diabetic and diabetic obese patients examined versus control group. These results are compatible with Jia *et al*.

[28], who have revealed that plasma free fatty acids (FFAs) rise play a significant role of insulin resistance in the production of type 2 diabetes. Type 2 diabetes progresses, as pancreatic β-cells gradually do not generate sufficient insulin to accommodate persistent insulin resistance. Regarding insulin and insulin resistance, insulin was declined in diabetic and diabetic obese patients compared to control group.

Titov *et al*. [29] demonstrated that, in skeletal muscles, where most glucose absorption is induced by insulin, increased FFA levels have been determined to result in insulin resistance. Prior to that, it was assumed that FFA production from accumulated fat cells disrupted homeostasis of glucose through the cycle of Randle glucose fatty acid. The absorption of glucose is declined when FFA oxidation met the tissue energy needs. It was believed that FFAs oxidation resulted in reduced glucose oxidation and elevated levels of intracellular citrate that could minimize glycolysis and glucose absorption. Insulin-stimulating glucose began to absorb normally for several hours after the fatty acids were neglected for maximum carbohydrate oxidation. Additionally, limiting the absorption of fatty acids by directly suppressing the trans membrane fatty carrier CD36 to reduce the mitochondrial content of the muscle, at least partly, to decrease in LPL expression and consequently to a significant reduction in the activation of PPAR-gamma.

In addition, the accumulation of macrophages in white fat tissue characterizes the obesity. Adipose tissue macrophages (ATMs) may contribute to the adipokins production. A major role of ATMs in insulin resistance is correlated with obesity displays that inhibiting of macrophage recruitment in obesity improves insulin resistance [30].

Our study results are coping with the study of García-Chapa et al. [31] and Sampson *et al*. [32] who demonstrated that, the overall risk of atherosclerosis is exaggerated when elevated TAG coexists with atherogenic total cholesterol concentrations. The main drawback is the consequent elevation of LDL-C in the presence of T2DM. T2DMraises the circulating FFAs, leading to exaggeration of the formation of very low density lipoprotein cholesterol (VLDL-C). TAG-rich VLDL then exchanges TAG and cholesterol to LDL-C and HDL-C, inducing the synthesis of TG-rich, small, dense LDL and TAG-rich HDL clearance. T2DMrises FFAs resulting in an increased manufacturing of VLDL-C. Therefore, hyperglycemia can be correlated with proatherogenic changes in plasma lipoproteins.

Furthermore, Neeli *et al*. [33], reported that, diabetic dyslipidemia may occur due to the modulation of lipoprotein lipase activity resulting in insulin secretion and/or action defects that lead to a decrease in the circulation of chylomicrons and VLDL. Lipoprotein lipase, an enzyme sensitive to insulin, can be associated with endothelial vascular surface proteoglycans and mediate TAG hydrolysis in chylomicrons and VLDL to generate monoacylglycerol and FFAs. It also encourages the oxidation of adipocytes in FFA.

De Grooth *et al*. [34], revealed that, T2DM can increase the effect of cholesterol ester transfer protein (CETP), which is responsible for transferring of TAG from VLDL-C, IDL-C or LDL-C to cholesteryl esters. This results in lower levels of HDL-C and high levels of lipoprotein rich in TAG (chylomicrons and VLDL-C particles).

Kocks and Kritharides [35] study is in line with our study as the elevated synthesis of adipose tissue FFA, in particular visceral fat into the portal veins, facilitates the triglyceride-rich lipoproteins production in the liver by VLDL particles release. This caused a drop in HDL-C concentrations in other lipoprotein particles.

Our current study revealed a critical rise in irisin and leptin in diabetic obese patients in relation to diabetic non-obese patients as well as in healthy volunteers. Aydin [36] emphasized this; irisin converts white adipose tissue to brown adipose tissues. Besides, thermogenesis, energy consumption, weight reduction and finally homeostasis of glucose are controlled by a noticeable increase in the uncoupling protein 1 in dark colored fat. Irisin, likewise referred to

as "exercise hormone" is generated by FNDC5, the extracellular fraction of membrane protein type I, after the stimulation by PGC-1α.

The regulation of irisin in obesity can, however, be quite different. In extremely obese women there was an increase in circulating irisin concentrations. The mass of fat and body mass index (BMI) were connected with the irisin levels. Irisin was also tested as an indicator of adverse metabolic syndrome results. Higher levels of irisin in blood were correlated with lower HDL cholesterol in obese ones with different cardiovascular disorders. Irisin also has a positive relationship with VLDL, triacylglycerol and total cholesterol [37].

Sáinz et al. [38] stated that, serum leptin in obese rats is high and can lead to a rise in signaling of PGC-1α and also an elevation in FNDC5 mRNA articulation might be an independent physical exercise. A causal link was noted among leptin concentration, the expression of PGC-1α and FNDC5 mRNA articulation. Leptin is an arbiter between skeletal muscle and fat tissue, to elevate the energy consumption as well as fat reduction through irisin. Considering the leptin-resistant position especially, leptin and irisin both in the hippocampuscan affect the sign transducer and the signaling of activator of transcription 3.

On the other hand, our study showed that adiponectin was significantly decreased in diabetic obese patients as well as in diabetic non-obese patients in relation to healthy volunteers.

Kim et al. [39] study goes hand in hand with ours which revealed that, adiponectin in diabetic obese individuals and in diabetic non-obese individuals in relation to healthy volunteers has been significantly reduced. Adiponectin increases insulin release through the activation of insulin gene expression and the insulin particles exocytosis. In brain, adiponectin also increases the energy consumption and can lead to weight loss. Adiponectin is freely and adversely associated with metabolic syndrome (MetS), insulin obstruction, T2DM, body weight, blood pressure and plasma lipids.

PNPLA3 rs738409 GG-genotype is related tothe reduction of triacylglycerol and cholesterol concentrations in obese [40] or glucose uncharitable ones [41] as a result of lowered hydrolysis of triacylglycerol and intrahepatic fat aggregation caused by obesity [42]. In fact, PNPLA3 showed a significant hydrolysis for the three major glycerolipids and the PNPLA3 I148 M variants with a significantly reduced level for glycerolipids Vmax [43,44]. Reduction of PNPLA3 hydrolysis function by the I148 M variant was shown as: VLDL kinetics analysis following a bolus imbuement of stable isotopes in overweight/fat men confirmed that the activation of VLDL1 particles secretion was decreased, the [3H]—triacylglycerol hydrolysis average during lipid degradation reduced, as well as the I148 M variant improved the labeled triacylglycerol accumulation beside large amounts of FFAs [42].

In our study, The PNPLA3 rs738409 gene polymorphism frequency was highly significant in diabetic obese patients when compared to diabetic none obese group. This result goes with that of Karamfilova et al. [45] who demonstrated a relationship between diabetes mellitus and the PNPLA3I148M allele. Significantly higher percentage of CG and GG genotype was found in patients with insulin resistance. Furthermore, Cox et al. [46] reported that the PNPLA3 SNP rs738409 contributes to risk for increased liver fat content in African Americans with T2DM. This effect is independent from blood lipids and blood glucose.

According to US national library of Medicine (NIH), this research is the first trial to asses PNPLA3 rs738409 gene polymorphism frequency in diabetic obese patients. There is only one other study in chronic hepatitis C infection and liver fibrosis. The frequency of different genotypes in control group in our study is going almost in agreement with that of Youssef et al. [47] study as follows: CC genotype frequency is 36.7% and 35.8%, CG genotype frequency is 53.3 and 57.5 and GG genotype frequency is 9% and 6.7% respectively.

Our experiment demonstrated that the sequence variations in the ADIPOQ gene rs266729 G>C contribute to susceptibility to obesity in T2DM and it showed association with T2DM

(Table 4). Coping with our results, the variants of ADIPOQ gene rs266729 and the prevalence of diabetes was studied in a Jordanian study which revealed that a greater number of diabetic patients carried this polymorphism, suggesting its correlation with the risk of T2DM [48]. Additional Tunisian study has investigated 13 different polymorphisms of the ADIPOQ genes. Among the 13 variants tested, the same variant (rs266729) showed a strong association with T2DM [49]. Another Omani study identified the frequency of SNP of rs266729 and found that it is significantly correlated with body weight, waist circumference, BMI and percentage of total body fat [50].

Leptin SNPrs2167270 G>A study confirmed that, it is highly associated with obesity, since the leptin resistance is the prime characteristic or main feature of the obesity. Although this *LEP* SNP is located in non-coding exon, it was found to be associated with severe obesity and increased leptin levels. This may be attributed to its critical position in 5′UTR regulatory region [51]. Pawlik et al. [52] demonstrated an association between the *LEP* rs2167270 polymorphism and the requirement for daily insulin in diabetic women.

Both human and mice PNPLA3 have as of late shown to exhibit an acyl-CoA-dependent lysophosphatidic acid acyltransferase (LPAAT) action, which progresses cell lipid union by changing LPA to phosphatidic acid. High sucrose diet up-directs PNPLA3 of wild-type mice in liver, which have shown high LPAAT activity at the same time [53].

## 5. Conclusions

Irisin and leptin levels elevation can predict the insulin resistance in obese patients. *PNPLA3* I148 M allele carriers rs738409C>G, ADIPOQ gene rs266729 G>C and LEP gene rs2167270 G>A polymorphisms had higher susceptibility rate for DM in obese people than wild type. This could be considered as an adjustable response to counter the impact of obesity on glucose homeostasis.

## 6. Limitations of the study

This study was carried out on a small sample of type2 diabetes of Egyptian citizen. The allele frequencies are important for studying complex diseases such as obesity and diabetes. Gene polymorphisms associate significantly with certain diabetic phenotypes in a given ethnic group may be absent or rare in another ethnic group. Moreover, racial differences in the prevalence of certain allele could account for certain proportion of disease trait variation between different ethnicities. In fact, most allele frequencies in the databases were obtained from studies of relative small samples (dozens or so), which could yield large sampling errors. Selection of a right SNP for a study is rather difficult. The discrepancies in the SNP data may yield wrong conclusion about suitability of particular SNPs for studying genetic basis of these differences.

## 7. Future prospective

In future association studies, to obtain reliable estimates of allele frequency distributions, relatively larger sample sizes considering also environmental factor in urban and rural areas in Egypt are needed. A comparison of SNP allele frequencies among different ethnicities can provide valuable information for better mapping of disease association. Various individual studies are needed to check the accuracy and completeness of genotyping results and to compare the SNP allele frequencies in different ethnic groups. Studying more different genotyping sites is recommended.

## Supporting information

**S1 Data. Demographic data.**
(XLS)

## Acknowledgments

We thank the internal medicine department, Zagazig university hospital for helping in samples' collection.

## Author Contributions

**Methodology:** Omnia Aly, Hanan Hassan Zaki, Mohamed R. Herzalla, Ahmed Fathy, Nermin Raafat, Mohamed M. Hafez.

**Writing – original draft:** Omnia Aly, Hanan Hassan Zaki, Mohamed R. Herzalla, Ahmed Fathy, Nermin Raafat, Mohamed M. Hafez.

**Writing – review & editing:** Omnia Aly, Hanan Hassan Zaki, Mohamed R. Herzalla, Ahmed Fathy, Nermin Raafat, Mohamed M. Hafez.

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
