## [Editor Report · Decision Letter 0]

18 Dec 2019

PONE-D-19-31278

Gene Polymorphisms of Patatin-like phospholipase domain containing 3 (PNPLA3), Adiponectin, Leptin in Diabetic Obese Patients

PLOS ONE

Dear Dr Hafez,

Thank you for submitting your manuscript to PLOS ONE. After careful consideration, we feel that it has merit but does not fully meet PLOS ONE’s publication criteria as it currently stands. Therefore, we invite you to submit a revised version of the manuscript that addresses the points raised during the review process.

We would appreciate receiving your revised manuscript by Feb 01 2020 11:59PM. To enhance the reproducibility of your results, we recommend that if applicable you deposit your laboratory protocols in protocols.io, where a protocol can be assigned its own identifier (DOI) such that it can be cited independently in the future. For instructions see: http://journals.plos.org/plosone/s/submission-guidelines#loc-laboratory-protocols

We look forward to receiving your revised manuscript.

Kind regards,

Nadia M Hamdy, Ph.D.

Academic Editor

PLOS ONE

Additional Editor Comments:

Dear

English language needs revision and paragraphs need rephrasing as well

More details are required in the methods

Limitations and future prospective are to be there

Journal Requirements:

2. In the ethics statement in the manuscript and in the online submission form, please provide additional information about the patient records/samples used in your retrospective study. Specifically, please ensure that you have discussed whether all data/samples were fully anonymized before you accessed them and/or whether the IRB or ethics committee waived the requirement for informed consent. If patients provided informed written consent to have data/samples from their medical records used in research, please include this information.

3. Please include in your Methods section the date ranges over which you recruited participants to this study. In your Results, please also include a summary of patient demographics (eg. age, sex).

4. Your ethics statement must appear in the Methods section of your manuscript. If your ethics statement is written in any section besides the Methods, please move it to the Methods section and delete it from any other section. Please also ensure that your ethics statement is included in your manuscript, as the ethics section of your online submission will not be published alongside your manuscript.
---

## [Author Response · Author response to Decision Letter 0]

3 Feb 2020

Gene Polymorphisms of Patatin-like phospholipase domain containing 3 (PNPLA3), Adiponectin, Leptin in Diabetic Obese Patients

1. Comment: ʺEnglish language needs revision and paragraphs need rephrasing as wellʺ

Response: the whole manuscript starting from the title was revised by English native speaker and many paragraphs were rephrased or almost rewritten.

2. Comment: ʺMore details are required in the methodsʺ

Response: each protocol for each parameter was written as a separate item with separate title and the details for each technique was added including principle, steps, kit manufacturer and the machine used.

3. Comment: ʺLimitations are to be thereʺ

Response: a title of (Limitations of the study) was added after the conclusion section and all expected limitations were mentioned under this section.

4. Comment: ʺFuture prospective are to be thereʺ

Response: a title of (Future Prospective) was added after the limitations of the study section and all recommended future work was mentioned under this section.

*All changes done in the manuscript according to reviewer comments are highlighted in yellow and English language and grammar revisions are done but not highlighted as they are frequent.

---

## [Decision Letter · Decision Letter 1]

26 Mar 2020

PONE-D-19-31278R1

Gene Polymorphisms of Patatin-like phospholipase domain containing 3 (PNPLA3), Adiponectin, Leptin in Diabetic Obese Patients

PLOS ONE

Dear Dr Hafez,

Thank you for submitting your manuscript to PLOS ONE. After careful consideration, we feel that it has merit but does not fully meet PLOS ONE’s publication criteria as it currently stands. Therefore, we invite you to submit a revised version of the manuscript that addresses the points raised during the review process.

In addition to the Academic Editor's comments from the previous decision, we have now obtained an assessment from an external reviewer. Please find their comments included below.

We would appreciate receiving your revised manuscript by May 10 2020 11:59PM. To enhance the reproducibility of your results, we recommend that if applicable you deposit your laboratory protocols in protocols.io, where a protocol can be assigned its own identifier (DOI) such that it can be cited independently in the future. For instructions see: http://journals.plos.org/plosone/s/submission-guidelines#loc-laboratory-protocols

We look forward to receiving your revised manuscript.

Kind regards,

Artur Arikainen, Associate Editor, PLOS ONE

on behalf of

Nadia M Hamdy, Ph.D.

Academic Editor

PLOS ONE

Reviewers' comments:

Reviewer's Responses to Questions

**Comments to the Author**

1. If the authors have adequately addressed your comments raised in a previous round of review and you feel that this manuscript is now acceptable for publication, you may indicate that here to bypass the “Comments to the Author” section, enter your conflict of interest statement in the “Confidential to Editor” section, and submit your "Accept" recommendation.

Reviewer #1: (No Response)

2. Is the manuscript technically sound, and do the data support the conclusions?

Reviewer #1: (No Response)

3. Has the statistical analysis been performed appropriately and rigorously? 

Reviewer #1: (No Response)

4. Have the authors made all data underlying the findings in their manuscript fully available?

Reviewer #1: (No Response)

5. Is the manuscript presented in an intelligible fashion and written in standard English?

Reviewer #1: (No Response)

6. Review Comments to the Author

Reviewer #1: 1. English editing is highly recommended throughout the manuscript.

2. Abstract:

- Rephrase line 35-37.

3. Introduction:

- Add more references on PNPLA3, lane 76

4. Methods:

- “Fresh venous blood samples” explain what do you mean by “fresh” this is not an accurate word here, explain?? And rephase.

- Add references for PNPLA3 amplification and genotyping sections.

- A figure for PNPLA3 results should be added.

5. Results:

- include a table with the demographic data of all participants.

- Rephrase lines from 190-194 to be more simple and comprehensive.

-

6. Discussion:

- Authors should compare results of genotypes distribution of the genes with other studies in Egypt, specially in controls.

- Authors should compare polymorphism results of all genes with those of other studies.

7. Limitations of the study:

- State clearly that the small sample numbers is a limitation in the study of polymorphism of these genes in the herein study.

8.

7. PLOS authors have the option to publish the peer review history of their article (what does this mean?). If published, this will include your full peer review and any attached files.

Reviewer #1: No

---

## [Author Response · Author response to Decision Letter 1]

10 May 2020

Response to Reviewers

Gene Polymorphisms of Patatin-like phospholipase domain containing 3 (PNPLA3), Adiponectin, Leptin in Diabetic Obese Patients

1. Comment: English editing is highly recommended throughout the manuscript ʺ

Response: the whole manuscript was revised.

2. Comment: ʺAbstract: Rephrase line 35-37ʺ

Response: lines 34-37 were rephrased.

3. Comment: ʺIntroduction: Add more references on PNPLA3, line 76ʺ

Response: 3 more references were added, lines 78, 79.

4. Comment: ʺ Methods: Fresh venous blood samples” explain what do you mean by “fresh” this is not an accurate word here, explain?? And rephraseʺ

Response: it was written by mistake and it is modified into five ml

5. Comment: ʺ Methods: “Add references for PNPLA3 amplification and genotyping sections. ʺ

Response: the reference is added (Dutta, 2011) line 128 

6. Comment: ʺ Methods: “A figure for PNPLA3 results should be added.ʺ

Response: the gel electrophoresis pictures were added (figure 1 for PCR product at line 134) and (figure 2 for genotying, line 153).

7. Comment: ʺResults: include a table with the demographic data of all participants ʺ

Response: We put it at the supporting information files.

8. Comment: ʺResults: Rephrase lines from 190-194 to be more simple and comprehensive ʺ

Response: most of results paragraphs were made simpler and more comprehensive

9. Comment: ʺDiscussion: Authors should compare results of genotypes distribution of the genes with other studies in Egypt, especially in controls. 

Response: for PNPLA3 genotyping there is only one study was done in Egypt which was on Hepatitis C infection and liver fibrosis. We added citation for this study in discussion section with comparison between control groups of this study and our study. So this is the first time to assess PNPLA3 genotype frequency in diabetic obese patients in Egypt. ADIPOQ and LEP gene polymorphism which were done in Egypt were targeting other sites other than our target on ADIPOQ rs266729 and Leptin rs2167270. 

10. Comment: ʺ Discussion: Authors should compare polymorphism results of all genes with those of other studies ʺ

Response: Regarding PNPLA3 genotyping, we compared among our study and other different studies but there are very few studies which performed this polymorphism in diabetic obese patients but we mentioned them in discussion. Regarding ADIPOQ and LEP genotyping, were compared with different mentioned studies in the discussion with new citations. 

11. Comment: ʺLimitations of the study: State clearly that the small sample numbers is a limitation in the study of polymorphism of these genes in the herein studyʺ

Response: the small sample size limitation is stated and recommendations were restated.

*All changes done in the manuscript according to reviewer comments are highlighted in blue and English language and grammar revisions are done.

---

## [Editor Report · Decision Letter 2]

28 May 2020

Gene Polymorphisms of Patatin-like phospholipase domain containing 3 (PNPLA3), Adiponectin, Leptin in Diabetic Obese Patients

PONE-D-19-31278R2

Dear Dr. Hafez,

We are pleased to inform you that your manuscript has been judged scientifically suitable for publication and will be formally accepted for publication once it complies with all outstanding technical requirements.

With kind regards,

Nadia M Hamdy, Ph.D.

Academic Editor

PLOS ONE
---

## [Editor Report · Acceptance letter]

4 Jun 2020

PONE-D-19-31278R2 

Gene Polymorphisms of Patatin-like phospholipase domain containing 3 (PNPLA3), Adiponectin, Leptin in Diabetic Obese Patients 

Dear Dr. Hafez:

I'm pleased to inform you that your manuscript has been deemed suitable for publication in PLOS ONE. Congratulations! Your manuscript is now with our production department. 

Kind regards, 

on behalf of

Professor Nadia M Hamdy 

Academic Editor

PLOS ONE